# Effect of Extraction Solvent and Temperature on Polyphenol Profiles, Antioxidant and Anti-Inflammatory Effects of Red Grape Skin By-Product

**DOI:** 10.3390/molecules26185454

**Published:** 2021-09-07

**Authors:** Giovanna Baron, Giulio Ferrario, Cristina Marinello, Marina Carini, Paolo Morazzoni, Giancarlo Aldini

**Affiliations:** 1Department of Pharmaceutical Sciences (DISFARM), Università degli Studi di Milano, Via Mangiagalli 25, 20133 Milan, Italy; giulio.ferrario1@unimi.it (G.F.); cristina.marinello@unimi.it (C.M.); marina.carini@unimi.it (M.C.); giancarlo.aldini@unimi.it (G.A.); 2Divisione Nutraceutica, Distillerie Umberto Bonollo S.p.A, 35035 Mestrino, Italy; paolo.morazzoni@bonollo.it

**Keywords:** *Vitis vinifera*, by-products, polyphenols, high-resolution mass spectrometry, antioxidant, inflammation

## Abstract

A fully-detailed LC-MS qualitative profiling of red grape skin, extracted with a mixture of ethanol and water (70:30 *v*:*v*) has permitted the identification of 65 compounds which can be classified into the following chemical classes: organic and phenolic acids (14 compounds), stilbenoids (1 compound), flavanols (21 compounds), flavonols (15 compounds) and anthocyanins (14 compounds). The extraction yield obtained with water at different temperatures (100 °C, 70 °C, room temperature) was then evaluated and the overall polyphenol content indicates that EtOH:H_2_O solvent is the most efficient and selective for polyphenol extraction. However, by analyzing the recovery yield of each single polyphenol, we found that water extraction under heating conditions is effective (extraction yield similar or even better in respect to the binary solvent) for some polyphenolic classes, such as hydrophilic procyanidins, phenolic acids, flavonol glucosides and stilbenoids. However, according to their lipophilic character, a poor yield was found for the most lipophilic components, such as flavonol aglycones, and in general for anthocyanins. The radical scavenging activity was in accordance with the polyphenol content, and hence, much higher for the extract obtained with the binary solvent in respect to water extraction. All the tested extracts were found to have an anti-inflammatory activity in the R3/1 cell line with NF-kb reporter challenged with 0.01 µg/mL of IL-1α, in a 1 to 250 µg/mL concentration range. An intriguing result was that the EtOH:H_2_O extract was found to be superimposable with that obtained using water at 100 °C despite the lower polyphenol content. Taken together, the results show the bioactive potentialities of grape skin extracts and the possibility to exploit this rich industrial waste. Water extraction carried out by heating is an easy, low-cost and environmentally friendly extraction method for some polyphenol classes and may have great potential for extracts with anti-inflammatory activities.

## 1. Introduction

Grapes are amongst the most cultivated fruits in the world. In Europe alone, around 3.5 million hectares are dedicated to grape cultivation with a production of almost 27 million tons of fruit [1]. It is also estimated that 14.5 million tons of grape by-products are generated annually [2]. The main component of this waste is the grape pomace, mainly composed (50–65%) of grape skin [3]. Even though it is mostly used as compost or animal fertilizer, the phenolic-rich composition of the skin is what supports its use a source of bioactive phytochemicals. It contains anthocyanins alongside various members of the flavonoid family (flavan-3-ols, flavonols and flavanones), which have shown healthy activity as antioxidant and anti-inflammatory agents [4,5,6].

Hence, grapes skin represent a valuable source of bioactive polyphenols and would represent a valuable industrial source to satisfy the growing demand. A recent study conducted by Transparency Market Research, a global market intelligence group, has predicted a further increase in the polyphenol market owing to increasing demand and market size. This study indicates that the global demand for polyphenols in the global polyphenol market was valued at USD 761.9 million in 2020 and is expected to reach USD 969.2 million by the end of 2026, growing at a CAGR (Compound Annual Growth Rate) of 3.5% between 2021–2026 [7].

The appropriate industrial evaluation of red grape skin needs a suitable and green extraction method which should have the least experimental set-up, a low cost and have environmental and user-friendly characteristics.

Although many technological advancements in polyphenol extraction have been proposed, such as supercritical fluid extraction (SFE), ultrasound-assisted extraction (UAE), microwave assisted extraction (MAE), pressurized liquid extraction (PLE) and pressurized hot water extraction (PHWE) [8,9], the solid–liquid extraction (SLE), which simply consists of solvent application and leaching, remains the most popular. Since the structure of phenolic compounds determines their solubility in solvents of different polarity, the type of extraction solvent may have a significant impact on the yield of extraction polyphenols from plants material [10].

There are some reports concerning optimization of extraction conditions of the phenolic compound content and antioxidant activities of some plant foods, and the optimal procedure is usually different for different plant matrices and depends on the polyphenol composition [11,12,13]. For instance, acetone has been proven efficient in polyphenol extraction from lychee flowers compared to methanol, water and ethanol [14]. However, another study reported water as the best solvent for polyphenol extraction from walnut green husks [15]. In general, binary solvents such as ethyl acetate, acetone, methanol and ethanol with a differing water content are used as suitable extraction solvents of polyphenols from raw material.

However, legal limitations for solvent residues and restrictions on the use of conventional organic solvents are becoming more and more rigorous, especially in the fields of food and pharmaceutical, and together with the use of flammable and costly solvents, they represent a limit for industrial extraction.

In this work, a selected red grape skin by-product was extracted using water extraction at different temperatures and the metabolomic, antioxidant and anti-inflammatory profiles compared in respect to those obtained using EtOH:H_2_O 70:30 (% *v*:*v*). Our aim is to fully characterize the profile and activities of the polyphenol fractions isolated by a water-based extraction of red grape skin and evaluate the temperature effect thus to understand if this easy, low-cost and environmentally friendly extraction method may prompt industrial interest. Polyphenol profiling was carried out by targeted LC-MS analyses and the antioxidant and anti-inflammatory activities of the extracts tested by in vitro and cell methods, respectively.

## 2. Material and Methods

### 2.1. Chemicals

6-Hydroxy-2,5,7,8-tetramethyl-3,4-dihydrochromene-2-carboxylic acid (trolox), (−)-epicatechin, ethyl gallate, protocatechuic acid, Folin–Ciocalteu reagent, sodium carbonate, gallic acid, (+)-catechin, vanillin, hydrochloric acid, potassium chloride, sodium acetate, DMSO, 3-(4,5-Dimethyl-2-thiazolyl)-2,5-diphenyl-2H-tetrazolium bromide (MTT), IL1α, ethanol, formic acid and LC-MS grade solvents were purchased from Merck KGaA, Darmstadt, Germany. Quercetin 3-galactoside and malvidin 3-glucoside were obtained from Extrasynthese (Genay CEDEX, France). LC-grade H_2_O (18 MΩ cm) was prepared with a Milli-Q H_2_O purification system (Millipore, Bedford, MA, USA).

### 2.2. Sample Preparation

Extraction from dried red grape skin derived from controlled and selected wineries in central Italy was carried out under the following conditions: 2 g of raw material in 40 mL of EtOH-H_2_O 70:30 (% *v*/*v*) under magnetic stirring for 24 h at room temperature (extract A); 2 g of raw material in 40 mL of EtOH-H_2_O 70:30 (% *v*/*v*) at 50 °C for one hour followed by 24 h at room temperature under magnetic stirring (extract A50); 2 g of raw material in 40 mL of H_2_O (100%) at 100 °C for one hour followed by 24 h at room temperature under magnetic stirring (extract B); 2 g of raw material in 40 mL of H_2_O (100%) at 70 °C for one hour followed by 24 h at room temperature under magnetic stirring (extract C); 2 g of raw material in 40 mL of H_2_O (100%) under magnetic stirring for 24 h at room temperature (extract D).

Samples were then centrifuged at 5000× *g* for 10 min, filtered on 0.45 µm filters and dried overnight under a vacuum. The conditions used combined the idea of a low-cost and simple method easily transferrable to industrial scale. Room temperature was selected as the low-cost condition for 24 h, since it is demonstrated that a prolonged extraction period lead to the degradation of polyphenols [16]. A limited period of warming (1 h) was applied to avoid polyphenols degradation (especially anthocyanins [17]) at two different temperatures: boiling point (100 °C) and 70 °C, which is a mean of the most used temperature for water extract (60–80 °C) applied in the literature.

### 2.3. HPLC-HRMS Analysis

The extracts were resuspended in EtOH:H_2_O (50:50, % *v*/*v*) to obtain a concentration 5 × 10^4^ µg/mL, then diluted in H_2_O/HCOOH, 100/0.1% *v*/*v* (mobile phase A) at the final concentration of 5 × 10^3^ µg/mL and spiked with the internal standard (6-hydroxy-2,5,7,8-tetramethyl-3,4-dihydrochromene-2-carboxylic acid) at a final concentration of 5 × 10^−5^ M. The analyses were performed in triplicate by LC-HRMS: the mixtures were separated on a reversed-phase Agilent Zorbax SB-C18 column (150 × 2.1 mm, i.d. 3.5 µm, CPS analitica, Milan, Italy) by using a multi-step gradient of mobile phase A H_2_O-HCOOH (100:0.1, % *v*/*v*) and phase B CH_3_CN-HCOOH (100:0.1, % *v*/*v*) and analyzed by a LTQ Orbitrap XL mass spectrometer (Thermo Fisher Scientific, San Jose, CA, USA), as described by Baron et al. [18]. The spectra were acquired in negative and positive ion modes. Xcalibur 4.0 and Chromeleon Xpress 6.80 were used for instrument control and spectra analysis. A targeted data analysis was performed on the base of a database built searching in the literature for the known grape components [19,20,21,22,23,24,25,26,27,28] and the identification was carried out by using the exact mass (5 ppm of mass tolerance), the isotopic and fragmentation patterns. Quantitative analyses were assessed for epicatechin (0.11–7.26 µg/mL), quercetin 3-glucoside (0.045–2.90 µg/mL), ethyl gallate (0.019–4.95 µg/mL), protocatechuic acid (0.015–3.85 µg/mL) and malvidin 3-glucoside (0.048–12.3 µg/mL), using pure reference standards: calibration curves were built by plotting the peak area ratios of metabolite/trolox versus the nominal concentrations of the metabolite by weighted (1/x^2^) least-squares linear regression. Table 1 shows all the obtained linear curves and the relative limit of quantification (LOQ).

### 2.4. Determination of Total Polyphenol Content

Determination of total polyphenol content was conducted spectrophotometrically by a slight modification of the method described by Dewanto et al. [29]. Briefly, 12.5 µL of diluted extract (1000 µg/mL) was spiked with 50 µL of water, 12.5 µL of Folin–Ciocalteu reagent and 125 µL of Na_2_CO_3_ (7%) in a 96-well plate. The mixture was incubated at room temperature for 90 min in the dark, then the absorbance was read at 760 nm in a spectrophotometric microplate reader (BioTek’s PowerWave HT, Winooski, VT, USA). The results were then compared with a curve of gallic acid (0–1000 µg/mL) prepared with the same protocol of samples and expressed as % w/w (grams of polyphenols per one gram of starting material or dry extract).

### 2.5. Determination of Tannin Content

Determination of total tannin content was carried out by the vanillin assay [30] in 96-plate wells. Briefly, 150 µL of vanillin (4% solution in methanol) and 75 μL of concentrated HCl were added to 2.5 μL of diluted extract (1000 µg/mL). The mixture was mixed at room temperature for 15 min and then the absorbance measured against the blank at 500 nm using a microplate reader (BioTek’s PowerWave HT, Winooski, VT, USA). The readings were compared to standards containing known amounts of (+)-catechin (0–1000 µg/mL) and prepared with the same protocol as the samples. The results were then expressed as % w/w (grams of polyphenols per one gram of starting material or dry extract).

### 2.6. Total Anthocyanin Content

Total anthocyanin content was evaluated as cyanidin 3-glucoside at 520 nm, using a molar absorptivity coefficient of 26,900 M^−1^ cm^−1^ [31]. Extracts were diluted in buffer pH 1 (0.025 M KCl solution brought to pH 1 with 37% HCl) and pH 4.5 (0.4 M CH_3_COONa) and the absorbance read both at 520 nm and 700 nm in a spectrophotometric microplate reader after 15 min. Anthocyanin pigment concentration (as cyanidin 3-glucoside equivalents, µg/mL) was expressed as follows:(1)$$Anthocyanin\;pigment\;\left(µg/mL\right)=\frac{A\times MW\times DF\times 10^{3}}{\varepsilon \times l}$$
where *A* = (*A*_520nm_–*A*_700nm_) pH 1.0–(*A*_520nm_–*A*_700nm_) pH 4.5; *MW* (molecular weight) = 449.38 g/mol for cyanidin-3-glucoside; *DF* = dilution factor; *l* = pathlength in cm; *ε* = 26,900 molar extinction coefficient, in L mol^−1^ cm^−1^, for cyanidin 3-glucoside; and 10^3^ = factor for conversion from g to mg. The results were expressed as % w/w (grams of polyphenols per one gram of starting material or dry extract).

### 2.7. Antioxidant Activity

The antioxidant capacity was evaluated by the DPPH radical-scavenging method [32], with a few modifications. An aliquot of 100 µL of the extract solution at different concentrations (1–25 µg/mL) was spiked with 750 µL of ethanol and 400 µL of acetate buffer (0.1 M, pH 5.5), mixed and spiked with 250 µL of DPPH ethanolic solution (0.5 × 10^−3^ M). After 90 min at room temperature and in the dark the absorbance at 515 nm was measured for each sample analyzed in triplicate with a UV reader Shimadzu™ UV 1900 (Shimadzu, Milano, Italia). The percentage of inhibition was calculated as expressed by Equation (2) and the results expressed as mean ± SD.
(2)$$\frac{{\mathrm{Abs}}_{blank}-{\mathrm{Abs}}_{sample}}{{\mathrm{Abs}}_{blank}}\times 100$$

### 2.8. Anti-Inflammatory Activity

To assess the in vitro anti-inflammatory activity of the four different extracts, and evaluate the influence of the different extraction methods, the same cell model used by Baron et al. was deployed [18]. Extracts ranging from 1 to 250 µg/mL were incubated with R3/1 cells transduced with the NF-kb reporter gene and challenged with 0.01 µg/mL IL1α. The cell viability of the extracts on the same cell line was evaluated for the same concentrations used in the anti-inflammatory assay by MTT assay as reported by Baron et al. [18].

## 3. Results and Discussion

### 3.1. Absolute Quantification of Polyphenols, Tannins and Anthocyanins

As shown in Table 2, the % of dry extract in respect to the starting material was the highest for 100% water at 100 °C and reduced on the basis of the extraction temperature, reaching the lowest residue amount for EtOH:H_2_O and EtOH:H_2_O at 50 °C. By contrast, the extraction yield of polyphenols calculated in respect to the dry extract was much higher (almost 4 fold) in EtOH:H_2_O in respect to water extracts. The % yield of polyphenols in respect to the starting material was at 4% for EtOH:H_2_O and reduced by almost 50% in water and further reduced at 70 °C and room temperature. The data well indicate that EtOH:H_2_O solvent is the most efficient and selective for polyphenol extraction, while water is able to extract not only polyphenols but also many other constituents in a temperature dependent manner as already observed by González-Centeno et al. [33]. We observed no difference between the two hydroalcoholic extracts.

Table 3 reports the % of tannins and anthocyanins in respect to the dry extract and starting material. For EtOH:H_2_O, the % in respect to the starting material was 3.28 and 0.271, respectively, an amount almost two times and one and a half times higher in respect to water at 100 °C. The extraction yield of tannins and anthocyanins reduced on the basis of the temperature for the water extracts, while for the two hydroalcoholic extracts overlapped.

Comparing our results with previous studies [34,35,36] on grape skin extracts performed by SLE (Table 4), as expected we can generally observe a higher anthocyanins yield when acid is added in the extraction solvent and the temperature is lower, but their content is highly variable depending on the grape variety [35,36]. On the other hand, tannin yield seems not to be affected by acid or temperature. Observing the results, where a higher tannins/polyphenols ratio is found, there is a lower anthocyanins/polyphenols ratio due to the different physico-chemical properties of the two classes.

### 3.2. Compound Identification, Absolute and Relative Content

Figure 1 shows the LC-ESI-(-)-MS (Total Ion Current) profiles of the four extracts: extract A (panel A), extract B (panel B), extract C (panel C) and extract D (panel D). The comparison between the extract A and extract A50 profiles (which are almost overlapping) is reported in Appendix A. The peak eluting at 1.5 min and detected in all the extracts was assigned to tartaric acid (peak 1). The TIC showing the highest number of peaks is that referred to EtOH-H_2_O (panel A); eluted peaks of the extracts can be clustered into three main groups based on their RT: group (A), which includes peaks eluting between 3 and 20 min, group (B) medium lipophilic compounds, eluting between 20 and 40 min, and group (C) the most lipophilic compounds, eluting after 40 min. TICs relative to groups A, B and C are reported in Figure 2, Figure 3 and Figure 4, respectively. The LC-ESI-(+)-MS (positive ion mode) profiles are reported in Figure 5. The comparison between the extract A and extract A50 positive profiles (which are almost overlapping) are reported in Appendix A. Compound identification was carried out on the basis of the accurate MS and MS/MS fragmentation and the lists of identified peaks in all the extracts are reported in Table 5 (relative to the acquisition in negative ion mode) and Table 6 (relative to the acquisition in positive ion mode). The 65 identified compounds can be classified into the following chemical classes: organic and phenolic acids (14 compounds), stilbenoids (1 compound), flavanols (21 compounds), flavonols (15 compounds) and anthocyanins (14 compounds).

#### 3.2.1. Flavanols

As expected, most of the procyanidins (18 out of 21) eluted in group A, as the most hydrophilic compounds of the extract. A total number of 21 procyanidins were identified from monomers (catechin, epicatechin, gallocatechin, epigallocatechin, epicatechin–gallate) to oligomers, such as hexamer gallate.

The relative content of procyanidins calculated for the EtOH:H_2_O extract on the basis of the area of the peaks reconstituted by setting the [M-H]^−^ values as filter ion is summarized in Table 7. Dimers and the corresponding gallate esters, reaching more than 40% of the total procyanidins, represent the main species, followed by monomers (catechin, epicatechin and gallates), more than 25%, and trimers, with a relative content of 21%, followed by oligomers such as tetramer, pentamer and hexamer.

Procyanidins were present in all four extracts but with a different relative abundance as shown in Table 8. The relative percentage of each extract (where i = A or A50 or B or C or D) was calculated with the following formula with the area of the peaks identified in extract A set as 100%:(3)$$\frac{{\left({Area}_{analyte}/{Area}_{IS}\right)}_{extract\;i}}{{\left({Area}_{analyte}/{Area}_{IS}\right)}_{extract\;A}}\times 100$$

For some of the most hydrophilic compounds (procyanidins eluting up to 12.3 min) such as gallocatechin and epigallocatechin, the yield was much higher when the extraction process was carried out with water in respect to both EtOH:H_2_O A and A50. For other hydrophilic components such as procyanidin trimer, catechin, epicatechin and some procyanidins B, the extraction yield in water was slightly but significantly higher and this in accordance with their hydrophilicity. Heating at 50 °C seems not to globally affect the yield of extraction.

The extraction yield in water solvents was found reduced for procyanidins eluting from 12 to 19 min by almost 50% and much lower when the water-based extraction was carried out without a heating step which was also found to significantly increase the extraction of the most hydrophilic components.

In Table 9 are reported the absolute concentrations (µg/mL) of epicatechin in all the extracts.

#### 3.2.2. Flavonols

Flavonols were identified mainly as glycosides (8 out of 15), which eluted in group B. A glucuronide derivative (quercetin 3-glucuronide) and 6 aglycones (group C, except for myricetin) were also detected. Table 10 reports the relative abundance of flavonols in the EtOH:H_2_O. Quercetin is the main flavanol, accounting for almost 40%, followed by its glucuronide (16%), syringetin 3-glucoside (11%), myricetin (9%) and kaempferol (almost 6%). The other constituents have a relative content lower than 5%.

Table 11 shows the relative percentage in the five extracts. For all the identified flavonols both as aglycones and glucosides, the highest extraction yield was obtained when EtOH:H_2_O mixture was used as extraction solvent and the recovery yield followed the following order: extract B > extract C > extract D. By calculating the mean of %, the yield of extraction for the conjugated forms reduced to almost 66% when water was used in respect to EtOH:H_2_O and further decreased for aglycones which reduced by almost 88% in extract B and 94% in extract C to reach an almost negligible yield for H_2_O extraction in RT condition (0.3%). The extraction yield of EtOH:H_2_O after heating is comparable to that without heating both for glycosides and aglycones.

Table 12 reports the absolute concentrations of quercetin 3-galactoside in all the extracts.

#### 3.2.3. Stilbenoids, Phenolic and Organic Acids

Only one stilbenoid was detected in all the extracts, resveratrol glucoside, whose extraction yield was similar in extracts A, A50, B and C but reduced by almost 30% in extract D. A total number of 14 organic and phenolic acids were identified, whose relative abundance in extract A is reported in Table 13. Tartaric acid represents the main organic acid, accounting for more than 40%. Taking into consideration only phenolic acids, ethyl gallate is the main species (30.2%) followed by *p*-coumaroyl-glucosides (more than 20%) gallic acid (15.4%) and galloyl glucose (14.5%).

As shown in Table 14, some of the organic acids were extracted in a greater yield with water based solvents, such as protocatechuic acid, caftaric acid and *p*-coumaroyl hexoside 1, while for some others, such as galloyl glucose, protocatechuic acid hexoside, fertaric acid and ethyl gallate, the extraction yield was greater with EtOH:H_2_O (both extract A and A50). The % mean calculated not considering tartaric acid was higher by almost 40% for extract B in respect to A and reduced in C and D, this latter similar to A.

Table 15 reports the absolute concentrations of protocatechuic acid and ethyl gallate in all the extracts.

#### 3.2.4. Anthocyanins

As summarized in Table 6, two pyroanthocyanins (vitisin A and B) and twelve anthocyanins were identified: 5 glucosides, 1 caffeoyl glucoside, 2 acetyl glucosides and 4 coumaroyl glucosides. The relative content of anthocyanins in the EtOH:H_2_O extract is summarized in Table 16. Malvidin coumaroyl glucoside is the most abundant component, reaching a relative content of almost 55%, followed by malvidin glucoside (14%) and malvidin acetyl glucoside (almost 8%), and petunidin coumaroyl glucoside (6.4%) while the relative content of the other compounds was lower than 5%.

Malvidin coumaroyl glucoside was the most abundant also in the water extracts but its relative content is reduced in respect to malvidin acetyl glucoside as explained by its greater polarity.

The relative percentage of the aqueous extracts in respect to EtOH:H_2_O is reported in Table 17, showing that for most of the identified anthocyanins, the recovery yield followed the following order: extract A > extract A50 > extract B > extract C > extract D and the yields reduced proportionally with lipophilicity (RT). Vitisin B was an exception, since its extraction yield was greater in extracts C and D. By calculating the % mean, the anthocyanin content is almost 40% in B in respect to A, similar to C and reduced to 28% in D. Regarding the two hydroalcoholic extracts, heating reduces anthocyanins content of about 11%.

Table 18 reports the absolute concentrations of malvidin 3-glucoside in all the extracts.

Considering the almost overlapping analytical profile of extract A and A50, the activities were evaluated for the extract A only.

### 3.3. DPPH Assay

The radical scavenging activity was measured by the DPPH test as reported in Table 19. By calculating the activity as IC_50_ of the dry residue, as expected, extract A was significantly more active than water extracts, according to the polyphenol contents, which represent the main antioxidant components of the extract. The radical scavenging activity of the extracts was almost superimposable when the activity was normalized in respect to the polyphenol content demonstrating that the extracted polyphenols have a quite similar radical scavenging activity.

### 3.4. Anti-Inflammatory Activity

The MTT assay (Figure 6) indicates that cell viability was not affected by the four extracts at all concentrations tested and up to 250 μg/mL.

The anti-inflammatory activity was then tested by measuring the NF-kB dependent luciferase activity induced by IL1-α stimulus. The protective effect of the four extracts is reported as % of inhibition of luciferase activity in respect to control cells (Figure 7). For all the extracts, a dose-dependent effect was observed. All the data were statistically significant (*p* < 0.01) except for the 1 and 10 µg/mL concentrations of extract C and D and 10 µg/mL for extract A. The anti-inflammatory activity of extract A was found superimposable to B and this latter higher than C and D. The order of anti-inflammatory potency for the water extracts was expected on the basis of the content of polyphenols which, in line of principle, could be considered as the main anti-inflammatory components. However, the superimposable activity of A with B cannot be explained on the basis of the polyphenol content since it is more than three times higher in A in respect to B. This apparent contradiction can be explained in different ways. The first, is that the extracted polyphenols do not have the same anti-inflammatory potency and that those extracted in a similar way in A and B, such as the hydrophilic procyanidins, are the most effective anti-inflammatory components. A second explanation is that beside polyphenols, other constituents present in the water extracts but not present or present to a lesser extent in A, do have an anti-inflammatory activity which adds up to those of the polyphenols. A third explanation is that the water-soluble constituents such as macromolecules (proteins, polysaccharides) significantly influence polyphenol bioaccessibility in the cells [37]. Regarding this latter aspect it could be considered that water extraction could preserve the extracellular vesicles (EVs) which are naturally loaded with polyphenols and which could facilitate the cell absorption of polyphenols [38]. Water extraction but presumably not EtOH-H_2_O, would preserve the integrity of such vesicles.

## 4. Conclusions

In conclusion, a fully detailed qualitative profiling of red grape skin extracted with ethanol and water mixture has permitted the identification of 65 compounds which can be classified into the following chemical classes: organic and phenolic acids (14 compounds), stilbenoids (1 compound), flavanols (21 compounds), flavonols (15 compounds) and anthocyanins (14 compounds). The extraction yield of water at different temperatures (100 °C, 70 °C, room temperature) was then evaluated and the results indicate that extraction using water under heating conditions is effective in extracting some polyphenolic classes with a good recovery in respect to EtOH based binary extraction. In particular, a good extraction yield was observed for the hydrophilic procyanidins, phenolic acids, flavonol glucoside and stilbenoid. However, according to their lipophilic character, poor absorption was observed for the most lipophilic components, such as flavonol aglycones and in general for anthocyanins. The radical scavenging activity was in accordance with the polyphenol content and hence much higher for the extract obtained with the binary solvent in respect to water extraction. All the tested extracts were found effective as anti-inflammatory compounds, and an intriguing result was that the EtOH:H_2_O extract was found superimposable with that obtained using water at 100 °C. Taken together, the results indicate that water extraction carried out with heating condition is an easy, low-cost and environmentally friendly extraction method for some polyphenol classes and may have an industrial application for extracts with anti-inflammatory activity.

## Figures and Tables

**Figure 1 molecules-26-05454-f001:**
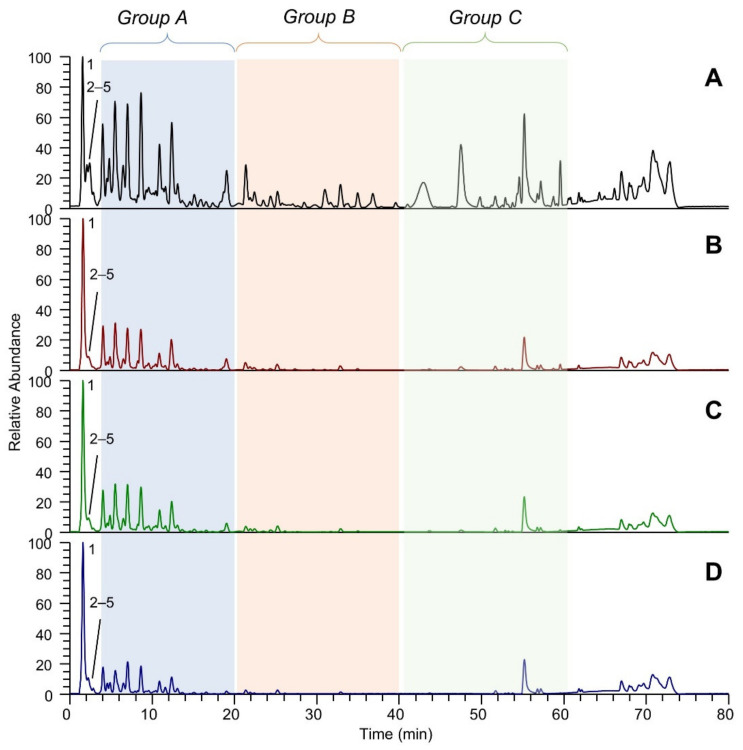
LC-ESI-(-)-MS total ion currents (TICs) in negative ion mode of the four extracts: (**A**) EtOH-H_2_O 70:30 (% *v*/*v*); extract A; (**B**) water at 100 °C, extract B; (**C**) water at 70 °C extract C; (**D**) water at room temperature, extract D. Peaks numbered from 1 to 5 eluted within 3 min and are not included in the groups A, B and C which are magnified in Figure 2, Figure 3 and Figure 4.

**Figure 2 molecules-26-05454-f002:**
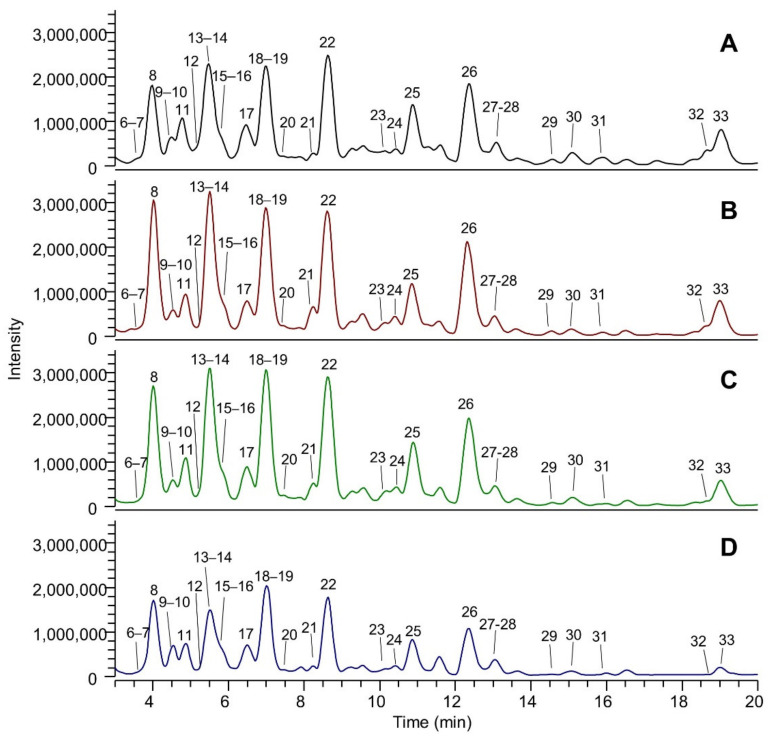
Magnification (range time 3–20 min) of the LC-ESI-(-)-MS total ion currents (TICs) recorded in negative ion mode of the four extracts: (**A**) EtOH-H_2_O 70:30 (% *v*/*v*); extract A; (**B**) water at 100 °C, extract B; (**C**) water at 70 °C extract C; (**D**) water at room temperature, extract D. Identified peaks (group A) are numbered progressively based on the RT and their assignment is reported in Table 5.

**Figure 3 molecules-26-05454-f003:**
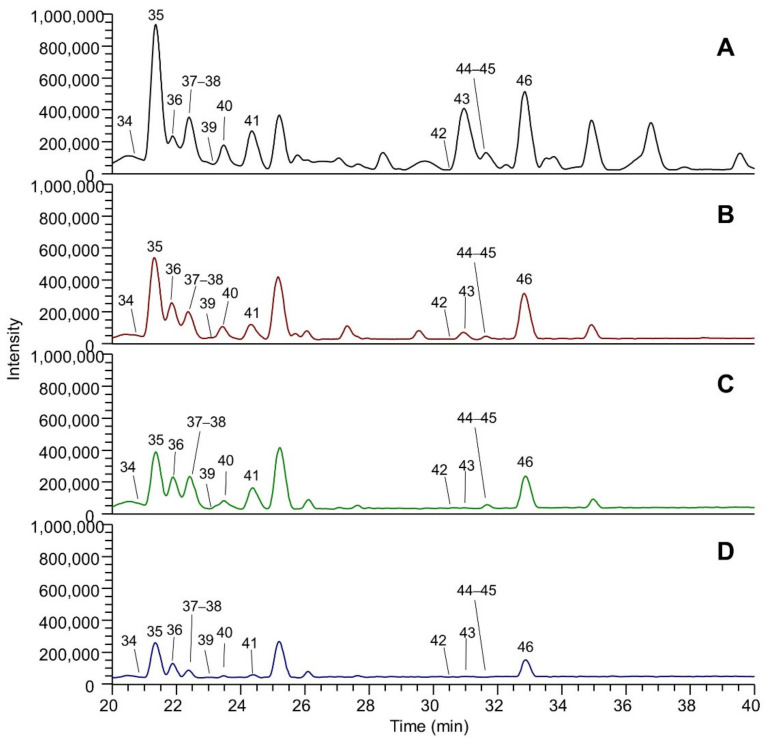
Magnification (range time 20–40 min) of the LC-ESI-(-)-MS total ion currents (TICs) recorded in negative ion mode of the four extracts: (**A**) EtOH-H_2_O 70:30 (% *v*/*v*); extract A; (**B**) water at 100 °C, extract B; (**C**) water at 70 °C extract C; (**D**) water at room temperature, extract D. Identified peaks (group B) are numbered progressively based on the RT and their assignment is reported in Table 5.

**Figure 4 molecules-26-05454-f004:**
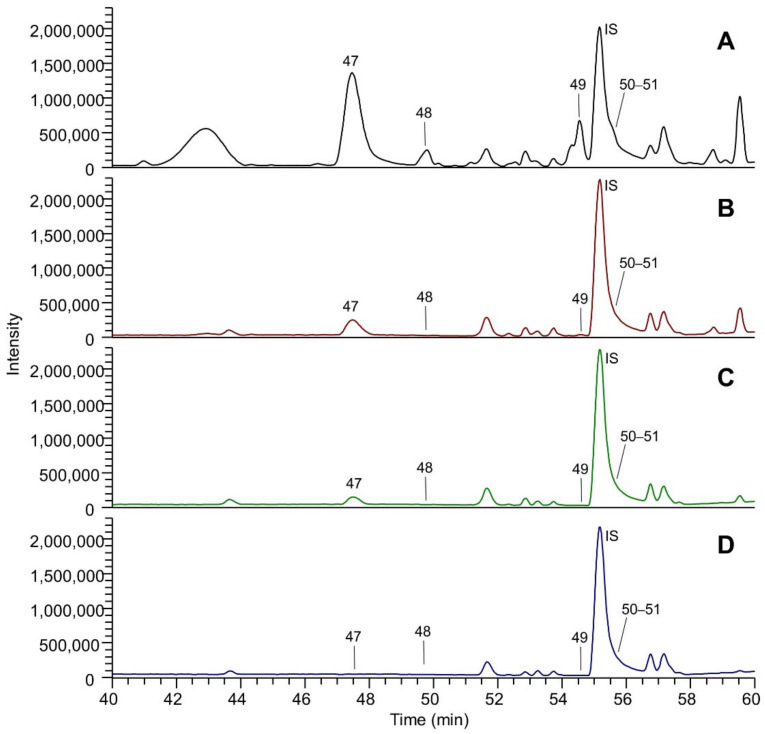
Magnification (range time 40–60 min) of the LC-ESI-(-)-MS total ion currents (TICs) recorded in negative ion mode of the four extracts: (**A**) EtOH-H_2_O 70:30 (% *v*/*v*); extract A; (**B**) water at 100 °C, extract B; (**C**) water at 70 °C extract C; (**D**) water at room temperature, extract D. Identified peaks (group C) are numbered progressively based on the RT and their assignment is reported in Table 5.

**Figure 5 molecules-26-05454-f005:**
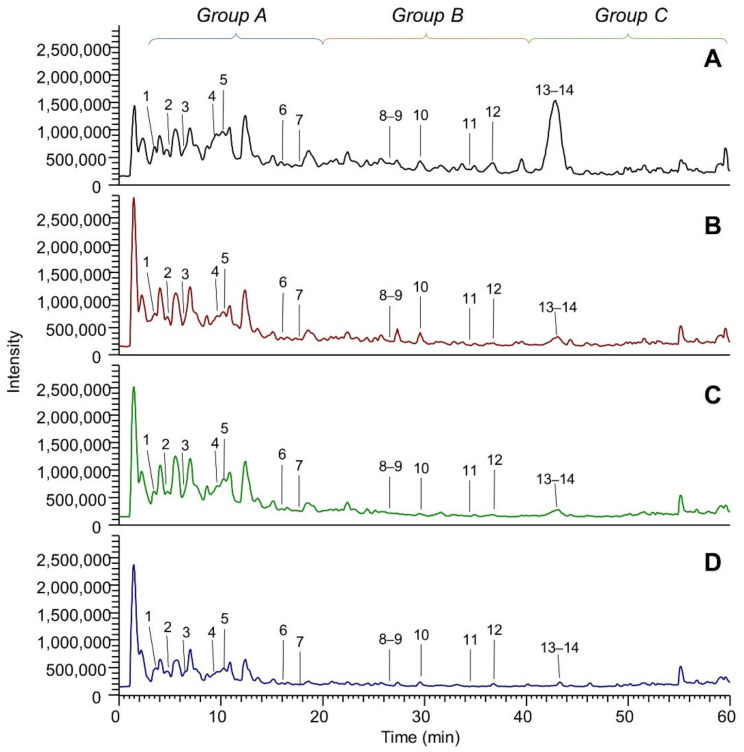
LC-ESI-(+)-MS total ion currents (TICs) in positive ion mode of the four extracts: (**A**) EtOH-H_2_O 70:30 (% *v*/*v*); extract A; (**B**) water at 100 °C, extract B; (**C**) water at 70 °C extract C; (**D**) water at room temperature, extract D. Peaks are numbered on the basis of the RT and the peak assignment is reported in Table 6.

**Figure 6 molecules-26-05454-f006:**
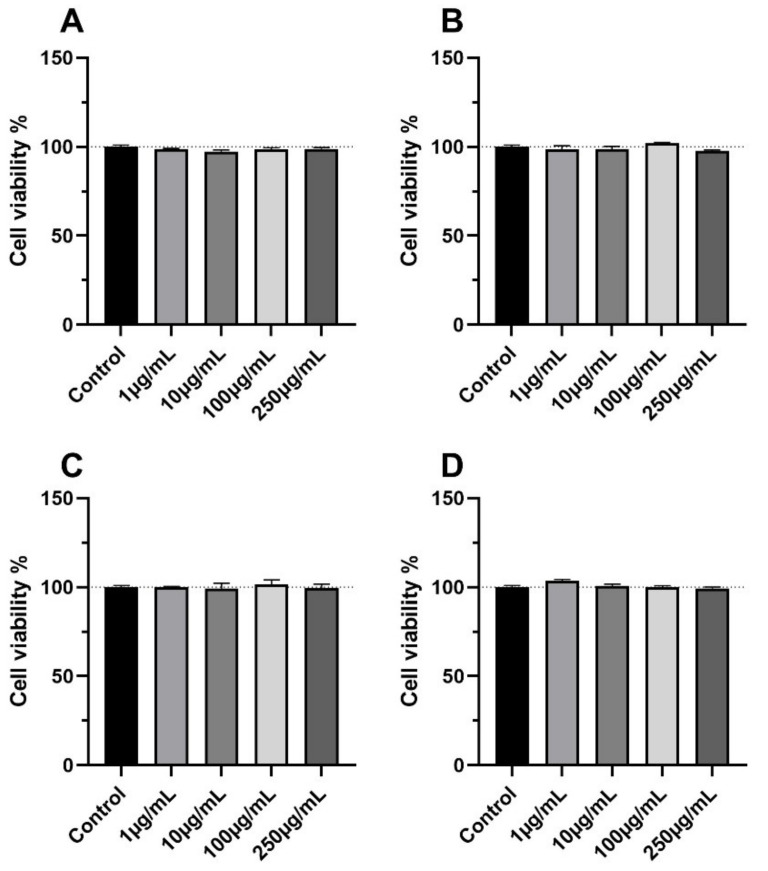
Viability of R3/1 cells incubated with the four extracts in a 1 to 250 µg/mL concentration range: (**A**) EtOH-H_2_O 70:30 (% *v*/*v*); extract A; (**B**) water at 100 °C, extract B; (**C**) water at 70 °C extract C; (**D**) water at room temperature, extract D. Cell viability was measured by the MTT assay. All the extracts were found not to significantly change the cell viability at all the tested concentrations. Statistical significance was calculated by ANOVA analysis followed by Dunnett’s multiple comparisons. A *p* > 0.05 was found for all the extracts.

**Figure 7 molecules-26-05454-f007:**
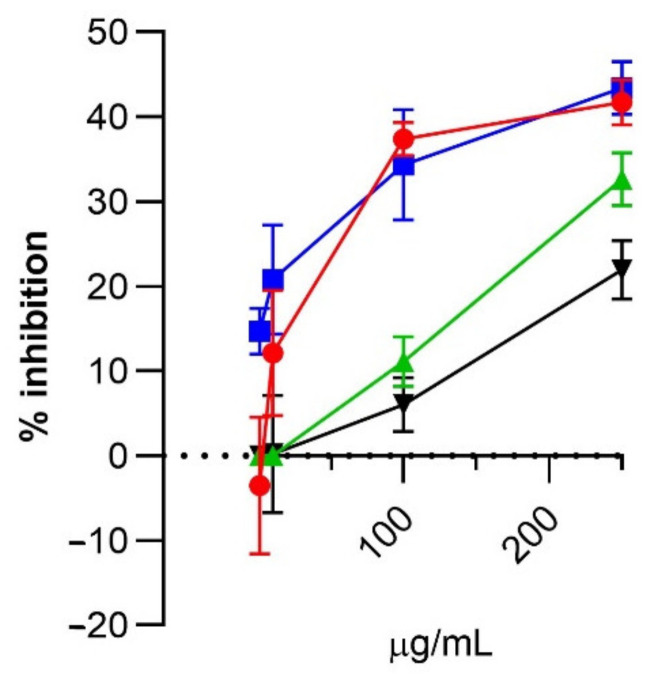
Dose-dependent anti-inflammatory activity of the four extracts. Extracts ranging from 1 to 250 µg/mL were incubated with R3/1 cells transduced with the NF-kb reporter gene and challenged with 0.01 µg/mL of IL1α. Results are reported as % inhibition of luciferase signal in respect to control cells. Symbols: ● extract A; ◼ extract B; ▲ extract C; ▼ extract D. Statistical analysis: ANOVA followed by Dunnett’s multiple comparisons test. The % inhibition found for each concentration was compared in respect to the control (untreated sample).

**Table 1 molecules-26-05454-t001:** Calibration curve parameters for the available compounds.

	Slope	Intercept	R^2^	LOQ
Epicatechin	0.3264	0.0287	0.9996	0.113
Quercetin 3-galactoside	0.5361	0.0139	0.9998	0.045
Ethyl gallate	0.3767	0.0194	0.9962	0.019
Protocatechuic acid	0.1417	0.0118	0.9916	0.015
Malvidin 3-glucoside	2.0151	0.3377	0.9965	0.048

**Table 2 molecules-26-05454-t002:** Total polyphenol content calculated in respect to the dry extract and to the starting material.

Extract	Dry Extract (% in Respect to Starting Material)	% Polyphenols (in Respect to Dry Extract)	% Polyphenols (in Respect to Starting Material)
Extract A	15.27 ± 0.38	28.23 ± 0.30	4.34 ± 0.07
Extract A50	12.50 ± 0.42	27.86 ± 0.17	4.15 ± 0.05
Extract B	35.00 ± 0.27	7.79 ± 0.10	2.77 ± 0.07
Extract C	28.81 ± 0.30	6.79 ± 0.17	1.92 ± 0.09
Extract D	21.69 ± 0.20	4.14 ± 0.07	0.83 ± 0.04

**Table 3 molecules-26-05454-t003:** Tannins and anthocyanins content calculated in respect to the dry extracts and to the starting material.

Extract	% Tannins (in Respect to Dry Extract)	% Tannins (in Respect to Starting Material)	% Anthocyanins (in Respect to Dry Extract)	% Anthocyanins (in Respect to Starting Material)
Extract A	21.49 ± 0.89	3.28 ± 0.14	1.78 ± 0.22	0.27 ± 0.03
Extract A50	21.52 ± 1.04	3.44 ± 0.17	1.52 ± 0.36	0.23 ± 0.06
Extract B	4.49 ± 0.32	1.57 ± 0.11	0.52 ± 0.10	0.18 ± 0.04
Extract C	3.78 ± 0.31	1.09 ± 0.09	0.45 ± 0.03	0.13 ± 0.01
Extract D	2.267 ± 0.22	0.49 ± 0.04	0.31 ± 0.06	0.07 ± 0.01

**Table 4 molecules-26-05454-t004:** Comparison of different SLE of grape skin extracts with relative tannins/polyphenols and anthocyanins/polyphenols ratios.

Starting Material	Extraction Solvent	Sample to Solvent Ratio	Time	Temperature	Tannins/ Polyphenols	Anthocyanins/ Polyphenols	Reference
Red grape skin without treatment	70% EtOH	2 g/40 mL	24 h	RT	0.750	0.081	Extract A
Red grape skin without treatment	70% EtOH	2 g/40 mL	1 h + 24 h	50 °C + RT	0.755	0.071	Extract A50
Red grape skin without treatment	H_2_O	2 g/40 mL	1 h + 24 h	100 °C + RT	0.571	0.125	Extract B
Red grape skin without treatment	H_2_O	2 g/40 mL	1 h + 24 h	70 °C + RT	0.500	0.133	Extract C
Red grape skin without treatment	H_2_O	2 g/40 mL	24 h	RT	0.500	0.150	Extract D
Red grape skin without treatment	80% EtOH	500 g/500 mL	1 h	RT	-	0.136	[34]
Lyophilized powder	0.01% HCl in 80% aqueous MeOH	-	18 h	RT	0.074–0.131	1.250–5.326	[36]
Skin powder	MeOH/H_2_O/HCl (80/20/0.1)	100 mg/8 mL	2 h	RT	0.595–1.222	0.093–0.347	[35]

**Table 5 molecules-26-05454-t005:** LC-ESI-(-)-MS identification (negative ion mode) of the EtOH-H_2_O extract constituents. The peak numbers refer to those reported in Figure 1, Figure 2, Figure 3 and Figure 4.

Group	Peak	Name	RT	*m/z* _calc_	*m/z* _exp_	Δppm	MS/MS
	1	Tartaric acid	1.5	149.0086	149.0093	−4.698	87–103–131
	2	Gallic acid	2.3	169.0137	169.0143	−3.550	125
	3	Galloyl glucose	2.4	331.0665	331.0664	0.362	125–169
	4	Protocatechuic acid hexoside	2.7	315.0716	315.0715	0.317	153
	5	Gallocatechin	2.9	305.0661	305.0665	−1.246	125–165–179–219–221–261–287
**A**	6	Protocatechuic acid	3.7	153.0188	153.0195	−4.705	109
	7	Caftaric acid	3.8	311.0403	311.0404	−0.289	149–179
	8	Procyandin B peak 1	4.0	577.1346	577.1331	2.582	289–407–425–451–559
	9	Epigallocatechin	4.3	305.0661	305.0660	0.393	125–165–179–219–221–261–287
	10	Caffeoyl hexoside 1	4.4	341.0873	341.0869	1.026	179–221–251–281
	11	Procyanidin B peak 2	4.9	577.1346	577.1330	2.755	289–407–425–451–559
	12	Caffeoyl hexooside 2	5.2	341.0873	341.0872	0.147	179–221–251–281
	13	Procyanidin trimer peak 1	5.4	865.1980	865.1945	4.022	287–289–407–451–577–695–713–739
	14	Catechin	5.5	289.0712	289.0706	2.110	179–205–245
	15	Coutaric acid	5.8	295.0454	295.0452	0.644	163
	16	Procyanidin trimer peak 2	5.8	865.1980	865.1937	4.947	287–289–407–451–577–695–713–739
	17	Procyanidin B peak3	6.5	577.1346	577.1334	2.062	289–407–425–451–559
	18	*p*-Coumaroyl hexoside 1	6.9	325.0923	325.0921	0.738	163–235–265
	19	Procyanidin B peak4	7.0	577.1346	577.1331	2.582	289–407–425–451–559
	20	Fertaric acid	7.3	325.0560	325.0560	−0.154	193
	21	*p*-Coumaroyl hexoside 2	8.3	325.0923	325.0922	0.431	163–235–265
	22	Epicatechin	8.7	289.0712	289.0708	1.418	179–205–245
	23	Vanillic acid hexoside	10.0	329.0873	329.0869	1.064	167–191–314
	24	Procyanidin trimer peak 3	10.2	865.1980	865.1937	4.947	287–289–407–451–577–695–713–739
	25	Procyanidin trimer peak 4	10.9	865.1980	865.1951	3.329	287–289–407–451–577–695–713–739
	26	Procyanidin dimer gallate	12.4	729.1431	729.1435	−0.590	287–289–407–451–559–577
	27	Ethyl gallate	13.1	197.0450	197.0456	−3.045	169
	28	Procyanidin tetramer	13.1	1153.2544	1153.2501	3.720	-
	29	Myricetin 3-galactoside	14.5	479.0826	479.0819	1.378	317
	30	Procyanidin pentamer	15.1	720.1559	720.1569	−1.389	-
	31	Procyanidin hexamer gallate	15.9	940.1956	940.1914	4.499	-
	32	Procyanidin dimer 3,3′-di-O-gallate	18.7	881.1565	881.1530	3.972	289–407–559–577–711–729
	33	Epicatechin 3-gallate	18.9	441.0822	441.0812	2.199	169–289
**B**	34	Quercetin 3-glucoside	20.7	463.0877	463.0871	1.188	301
	35	Quercetin 3-glucuronide	21.3	477.0669	477.0661	1.698	301
	36	Quercetin 3-galactoside	22.1	463.0877	463.0869	1.620	301
	37	Dihydroquercetin 3-rhamnoside	22.3	449.1084	449.1077	1.514	285–303
	38	Procyanidin trimer gallate peak 1	22.4	1017.2034	1017.203	0.364	-
	39	Kaempferol 3-hexoside	23.2	447.0927	447.0919	1.856	285
	40	Laricitrin 3-galactoside	23.5	493.0982	493.0979	0.629	331
	41	Procyanidin tetramer gallate	24.4	652.1322	652.1312	1.602	-
	42	Dihydrokaempferol 3-rhamnoside	30.5	433.1135	433.1133	0.393	287–269
	43	Myricetin	30.9	317.0297	317.0296	0.442	137–151–179
	44	Resveratrol glucoside	31.4	389.1236	389.1234	0.617	227
	45	Procyanidin trimer gallate peak 2	31.6	1017.2089	1017.2042	4.620	-
	46	Syringetin 3-glucoside	32.8	507.1139	507.1130	1.696	345
**C**	47	Quercetin	47.5	301.0348	301.0348	0.100	151–179
	48	Laricitrin	49.7	331.0454	331.0450	1.178	151–179–316
	49	Kaempferol	54.6	285.0399	285.0400	−0.316	151
	50	Syringetin	55.4	345.0610	345.0606	1.275	315–330
	51	Isorhamnetin	55.5	315.0505	315.0503	0.571	300

**Table 6 molecules-26-05454-t006:** LC-ESI-(+)-MS identification (positive ion mode) of the EtOH-H_2_O extract constituents. The peak numbers refer to those shown in Figure 5.

Peak	Name	RT	*m*/*z*_calc_	*m*/*z*_exp_	Δppm	MS/MS
1	Delphinidin 3-glucoside	3.7	465.1033	465.1028	1.054	303
2	Cyanidin 3-glucoside	5.2	449.1084	449.1079	1.069	287
3	Petunidin 3-glucoside	6.1	479.1189	479.1179	2.171	317
4	Peonidin 3-glucoside	8.8	463.124	463.1234	1.360	301
5	Malvidin 3-glucoside	9.6	493.1346	493.1339	1.399	331
6	Vitisin A	15.9	561.1244	561.1238	1.105	399
7	Vitisin B	18.2	517.1346	517.1339	1.334	355
8	Peonidin 3-(6″-acetyl)-glucoside	26.2	505.1346	505.1340	1.168	301
9	Malvidin 3-(6″-acetyl)-glucoside	26.6	535.1452	535.1445	1.215	331
10	Delphinidin 3-(6″-coumaroyl)-glucoside	29.6	611.1401	611.1396	0.769	303
11	Malvidin 3-(6″-caffeoyl)-glucoside	34.4	655.1663	655.1661	0.275	331
12	Petunidin 3-(6″-coumaroyl)-glucoside	36.4	625.1557	625.1551	0.992	317
13	Peonidin 3-(6″-coumaroyl)-glucoside	42.7	609.1608	609.1602	0.985	301
14	Malvidin 3-(6″-coumaroyl)-glucoside	42.9	639.1714	639.1706	1.205	331

**Table 7 molecules-26-05454-t007:** Relative content of flavanols in the EtOH-H_2_O extract. The % was determined by measuring the peak area of each compound in respect to the sum of the peak areas of all the identified flavanols.

Compound	Relative Content (%)
Gallocatechin	0.090 (±0.001)
Procyandin B peak1	6.46 (±0.05)
Epigallocatechin	0.0124 (±0.0001)
Procyanidin B peak2	4.50 (±0.07)
Procyanidin trimer peak 1	3.54 (±0.04)
Catechin	8.65 (±0.14)
Procyanidin trimer peak 2	2.46 (±0.04)
Procyanidin B peak3	6.11 (±0.06)
Procyanidin B peak4	9.75 (±0.20)
Epicatechin	11.11 (±0.20)
Procyanidin trimer peak 3	1.63 (±0.03)
Procyanidin trimer peak 4	8.51 (±0.12)
Dimer gallate	15.24 (±0.18)
Tetramer	1.85 (±0.02)
Pentamer	3.07 (±0.04)
Hexamer gallate	0.359 (±0.003)
Procyanidin dimer 3,3′-di-O-gallate	2.40 (±0.02)
epicatechin 3-gallate	5.75 (±0.03)
Trimer gallate	3.37 (±0.02)
Tetramer gallate	3.54 (±0.03)
Trimer gallate	1.60 (±0.02)

**Table 8 molecules-26-05454-t008:** Relative percentage (mean ± SD) of each flavanol in extracts A50, B, C and D calculated in respect to the content determined in extract A and set as 100%. The relative percentages were calculated by measuring the areas of the peaks reconstituted by setting the [M-H]^−^ as filter ions.

Compound	Extract A	Extract A50	Extract B	Extract C	Extract D
Flavanols					
Gallocatechin	100.00 (±1.24)	136.82 (±1.76)	255.85 (±1.33)	216.64 (±0.58)	20.72 (±1.24)
Procyandin B peak1	100.00 (±1.01)	104.22 (±0.70)	148.13 (±1.72)	137.17 (±1.74)	101.91 (±0.56)
Epigallocatechin	100.00 (±1.59)	143.61 (±1.67)	627.16 (±3.17)	660.25 (±2.00)	95.90 (±0.99)
Procyanidin B peak2	100.00 (±1.30)	106.07 (±0.95)	93.56 (±0.92)	106.66 (±0.98)	76.84 (±0.88)
Procyanidin trimer peak 1	100.00 (±0.93)	133.64 (±1.07)	88.95 (±0.72)	108.11 (±1.05)	57.64 (±0.74)
Catechin	100.00 (±1.83)	119.26 (±4.27)	127.26 (±1.39)	111.38 (±1.78)	66.85 (±0.17)
Procyanidin trimer peak 2	100.00 (±2.01)	94.01 (±1.68)	118.11 (±1.33)	122.07 (±1.34)	105.38 (±0.90)
Procyanidin B peak3	100.00 (±0.79)	91.52 (±0.84)	75.79 (±0.73)	89.42 (±0.62)	80.13 (±0.30)
Procyanidin B peak4	100.00 (±1.82)	108.22 (±1.53)	110.40 (±1.07)	122.58 (±0.67)	100.44 (±0.76)
Epicatechin	100.00 (±1.74)	106.68 (±1.32)	113.32 (±1.66)	120.78 (±1.67)	83.55 (±0.78)
Procyanidin trimer peak 3	100.00 (±1.56)	118.38 (±2.82)	104.93 (±1.05)	122.54 (±1.20)	70.62 (±1.15)
Procyanidin trimer peak 4	100.00 (±1.50)	120.04 (±0.71)	81.74 (±0.51)	110.56 (±1.02)	66.48 (±0.35)
Procyanidin dimer gallate	100.00 (±1.37)	93.17 (±0.78)	87.56 (±0.69)	81.17 (±0.51)	49.09 (±0.48)
Procyanidin tetramer	100.00 (±1.02)	93.26 (±0.34)	63.35 (±0.99)	92.90 (±1.43)	62.22 (±0.90)
Procyanidin pentamer	100.00 (±0.97)	116.02 (±0.97)	51.53 (±0.88)	70.97 (±0.73)	52.04 (±0.36)
Procyanidin hexamer gallate	100.00 (±0.89)	89.62 (±0.89)	63.56 (±0.64)	46.22 (±0.52)	47.44 (±0.81)
Procyanidin dimer 3,3′-di-O-gallate	100.00 (±0.93)	109.65 (±0.25)	65.90 (±0.59)	39.13 (±0.53)	12.70 (±0.07)
Epicatechin 3-gallate	100.00 (±0.62)	145.58 (±1.25)	102.69 (±0.41)	73.58 (±0.25)	29.99 (±0.38)
Procyanidin trimer gallate	100.00 (±0.86)	93.74 (±1.81)	54.75 (±0.78)	64.27 (±0.93)	28.62 (±0.59)
Procyanidin tetramer gallate	100.00 (±1.03)	98.70 (±1.09)	37.36 (±0.83)	46.61 (±0.65)	18.93 (±0.35)
Procyanidin trimer gallate	100.00 (±1.13)	117.58 (±0.71)	40.41 (±0.89)	38.39 (±0.17)	16.19 (±0.07)
MEAN	100	111	120	123	59

**Table 9 molecules-26-05454-t009:** Epicatechin absolute concentrations (µg/mL, mean ± SD) in the extracts.

	Extract A	Extract A50	Extract B	Extract C	Extract D
Epicatechin	3.83 (±0.07)	4.06 (±0.05)	4.42 (±0.01)	4.72 (±0.15)	3.24 (±0.03)

**Table 10 molecules-26-05454-t010:** Relative content of flavonols in the EtOH-H_2_O extract. The % was determined by measuring the peak area of each compound in respect to the sum of the peak areas of all the identified flavonols.

Compound	Relative Content (%)
Myricetin 3-galactoside	1.82 (±0.03)
Quercetin 3-glucoside	0.675 (±0.008)
Quercetin 3-glucuronide	16.33 (±0.22)
Quercetin 3-galactoside	2.01 (±0.05)
Dihydroquercetin 3-rhamnoside	1.58 (±0.02)
Kaempferol 3-hexoside	0.662 (±0.004)
Laricitrin 3-galactoside	3.13 (±0.03)
Dihydrokaempferol 3-rhamnoside	0.299 (±0.002)
Myricetin	9.07 (±0.05)
Syringetin 3-glucoside	11.48 (±0.07)
Quercetin	39.99 (±0.21)
Laricitrin	2.54 (±0.03)
Kaempferol	5.91 (±0.13)
Syringetin	1.08 (±0.01)
Isorhamnetin	3.41 (±0.03)

**Table 11 molecules-26-05454-t011:** Relative percentage (mean ± SD) of each flavonol in extracts A50, B, C and D calculated in respect to the content determined in extract A and set as 100%. The relative percentages were calculated by measuring the areas of the peaks reconstituted by setting the [M-H]^−^ as filter ions.

Compound	Extract A	Extract A50	Extract B	Extract C	Extract D
Flavonols					
Myricetin 3-galactoside	100.00 (±1.21)	115.48 (±1.46)	88.27 (±1.15)	66.97 (±1.14)	48.93 (±0.27)
Quercetin 3-glucoside	100.00 (±0.93)	117.94 (±1.63)	57.51 (±0.65)	42.33 (±0.40)	15.83 (±0.43)
Quercetin 3-glucuronide	100.00 (±1.64)	106.20 (±1.69)	56.28 (±0.33)	45.02 (±0.21)	37.60 (±0.23)
Quercetin 3-galactoside	100.00 (±2.26)	106.01 (±1.41)	75.67 (±1.81)	64.18 (±0.87)	34.22 (±1.30)
Dihydroquercetin 3-rhamnoside	100.00 (±1.10)	103.02 (±1.52)	77.01 (±0.56)	78.56 (±0.23)	109.37 (±1.19)
Kaempferol 3-hexoside	100.00 (±0.69)	102.09 (±0.54)	27.44 (±0.50)	10.47 (±0.11)	3.31 (±0.07)
Laricitrin 3-galactoside	100.00 (±1.20)	95.08 (±1.29)	63.05 (±0.97)	45.80 (±1.05)	34.88 (±0.24)
Dihydrokaempferol 3-rhamnoside	100.00 (±0.64)	97.02 (±0.34)	77.75 (±1.11)	91.09 (±0.63)	81.70 (±1.40)
Syringetin 3-glucoside	100.00 (±0.67)	104.41 (±0.80)	67.60 (±0.47)	53.43 (±0.33)	35.36 (±0.29)
Mean glycosides	100	101	66	55	45
Myricetin	100.00 (±0.57)	88.19 (±0.10)	18.06 (±0.18)	9.76 (±0.05)	0.115 (±0.004)
Quercetin	100.00 (±0.61)	89.55 (±0.74)	23.07 (±0.37)	13.88 (±0.29)	0.560 (±0.011)
Laricitrin	100.00 (±1.19)	99.98 (±2.74)	14.82 (±0.52)	6.21 (±0.06)	0.105 (±0.005)
Kaempferol	100.00 (±2.03)	102.49 (±1.10)	7.49 (±0.04)	3.61 (±0.07)	0.360 (±0.012)
Syringetin	100.00 (±0.67)	117.56 (±0.32)	5.11 (±0.34)	1.61 (±0.05)	0.364 (±0.080)
Isorhamnetin	100.00 (±0.73)	108.53 (±0.69)	5.96 (±0.62)	2.48 (±0.04)	0.170 (±0.021)
Mean aglycones	100	105	12	6	0.3

**Table 12 molecules-26-05454-t012:** Quercetin 3-galactoside absolute concentrations (µg/mL, mean ± SD) in the extracts.

	Extract A	Extract A50	Extract B	Extract C	Extract D
Quercetin 3-galactoside	0.251 (±0.006)	0.267 (±0.004)	0.183 (±0.0004)	0.152 (±0.003)	0.069 (±0.002)

**Table 13 molecules-26-05454-t013:** Relative content of phenolic acids in the EtOH-H_2_O extract. The % was determined by measuring the peak area of each compound in respect to the sum of the peak areas of all the identified phenolic acids.

Compound	Relative Content (%)
Gallic acid	15.40 (±0.12)
Galloyl glucose	14.48 (±0.18)
Protocatechuic acid hexoside	1.84 (±0.01)
Protocatechuic acid	1.24 (±0.02)
Caftaric acid	2.73 (±0.06)
Caffeoyl hexoside 1	3.76 (±0.04)
Caffeoyl hexoside 2	2.33 (±0.04)
Coutaric acid	1.24 (±0.02)
*p*-Coumaroyl glucoside 1	8.38 (±0.13)
Fertaric acid	3.54 (±0.04)
*p*-Coumaroyl-glucoside 2	12.3 (±0.30)
Vanillic acid hexoside	2.48 (±0.02)
Ethyl gallate	30.25 (±0.34)

**Table 14 molecules-26-05454-t014:** Relative percentage (mean ± SD) of each organic and phenolic acid in extracts A50 B, C and D calculated in respect to the content determined in extract A and set as 100%. The relative percentages were calculated by measuring the areas of the peaks reconstituted by setting the [M-H]^−^ as filter ions.

Compound	Extract A	Extract A50	Extract B	Extract C	Extract D
Organic and phenolic acids					
Tartaric acid	100.00 (±1.85)	36.79 (±0.33)	396.77 (±0.67)	361.47 (±0.77)	398.50 (±0.51)
Gallic acid	100.00 (±1.25)	105.17 (±1.29)	90.84 (±1.23)	93.47 (±0.83)	78.66 (±0.92)
Galloyl glucose	100.00 (±1.77)	84.24 (±0.92)	67.99 (±1.04)	71.69 (±1.05)	33.03 (±0.05)
Protocatechuic acid hexoside	100.00 (±0.59)	103.36 (±0.57)	73.76 (±0.84)	82.09 (±0.90)	80.16 (±0.94)
Protocatechuic acid	100.00 (±1.16)	61.67 (±0.39)	204.07 (±1.01)	196.38 (±1.03)	77.69 (±1.26)
Caftaric acid	100.00 (±1.63)	107.31 (±0.48)	244.63 (±2.09)	233.37 (±2.38)	225.71 (±0.45)
Caffeoyl hexoside 1	100.00 (±0.93)	102.34 (±0.69)	103.87 (±1.00)	106.29 (±0.65)	128.09 (±0.53)
Caffeoyl hexoside 2	100.00 (±0.72)	94.22 (±1.10)	181.01 (±0.87)	169.08 (±0.98)	165.44 (±0.58)
Coutaric acid	100.00 (±1.62)	116.31 (±2.12)	119.55 (±0.83)	108.56 (±0.53)	81.58 (±0.71)
*p*-Coumaroyl hexoside 1	100.00 (±1.11)	102.35 (±0.18)	214.77 (±1.03)	170.65 (±1.06)	82.63 (±1.19)
Fertaric acid	100.00 (±0.64)	98.56 (±1.42)	61.82 (±0.51)	64.50 (±0.45)	78.97 (±0.49)
*p*-Coumaroyl hexoside 2	100.00 (±1.93)	99.10 (±1.09)	219.64 (±1.62)	153.30 (±1.56)	93.92 (±1.51)
Vanillic acid hexoside	100.00 (±1.10)	105.65 (±0.38)	110.91 (±0.75)	105.28 (±0.83)	75.86 (±0.76)
Ethyl gallate	100.00 (±1.63)	100.00 (±0.58)	87.01 (±1.04)	84.00 (±0.20)	82.48 (±0.29)
Mean	100	94	136	126	99
Stilbenoids					
Resveratrol glucoside	100.00 (±2.11)	98.41 (±1.48)	84.71 (±1.91)	104.06 (±0.90)	72.66 (±1.00)

**Table 15 molecules-26-05454-t015:** Protocatechuic acid and ethyl gallate absolute concentrations (µg/mL, mean ± SD) in the extracts.

	Extract A	Extract A50	Extract B	Extract C	Extract D
Protocatechuic acid	0.681 (±0.009)	0.387 (±0.003)	1.47 (±0.01)	1.41 (±0.02)	0.509 (±0.003)
Ethyl gallate	1.71 (±0.01)	1.72 (±0.01)	1.48 (±0.02)	1.43 (±0.003)	1.40 (±0.005)

**Table 16 molecules-26-05454-t016:** Relative content of anthocyanins in the EtOH-H_2_O extract. The % was determined by measuring the peak area of each compound in respect to the sum of the peak areas of all the identified anthocyanins.

Compound	Relative Content (%)
Delphinidin 3-glucoside	0.410 (±0.012)
Cyanidin 3-glucoside	0.252 (±0.009)
Petunidin 3-glucoside	2.01 (±0.03)
Peonidin 3-glucoside	1.52 (±0.05)
Malvidin 3-glucoside	14.22 (±0.03)
Vitisin A	0.783 (±0.014)
Vitisin B	0.267 (±0.001)
Peonidin 3-(6″-acetyl)-glucoside	0.603 (±0.003)
Malvidin 3-(6″-acetyl)-glucoside	8.33 (±0.02)
Delphinidin 3-(6″-coumaroyl)-glucoside	2.62 (±0.01)
Malvidin 3-(6″-caffeoyl)-glucoside	3.40 (±0.07)
Petunidin 3-(6″-coumaroyl)-glucoside	5.99 (±0.11)
Peonidin 3-(6″-cis-coumaroyl)-glucoside	3.98 (±0.03)
Malvidin 3-(6″-cis-coumaroyl)-glucoside	55.61 (±0.16)

**Table 17 molecules-26-05454-t017:** Relative percentage (mean ± SD) of each anthocyanin in extracts A50, B, C and D calculated in respect to the content determined in extract A and set as 100%. The relative percentages were calculated by measuring the areas of the peaks reconstituted by setting the [M-H]^−^ as filter ions.

Compound	Extract A	Extract A50	Extract B	Extract C	Extract D
Anthocyanins					
Delphinidin 3-glucoside	100.00 (± 2.17)	89.59 (± 1.47)	73.16 (± 1.40)	76.02 (± 0.75)	21.82 (± 1.50)
Cyanidin 3-glucoside	100.00 (± 2.77)	79.57 (± 0.58)	50.64 (± 1.06)	30.19 (± 1.82)	27.58 (± 1.14)
Petunidin 3-glucoside	100.00 (± 2.19)	125.80 (± 1.74)	36.63 (± 0.14)	49.68 (± 0.38)	36.90 (± 2.83)
Peonidin 3-glucoside	100.00 (± 2.47)	80.24 (± 1.58)	52.65 (± 0.46)	54.12 (± 0.18)	37.96 (± 0.83)
Malvidin 3-glucoside	100.00 (± 0.82)	79.44 (± 1.70)	43.50 (± 0.46)	56.01 (± 0.62)	37.24 (± 0.42)
Vitisin A	100.00 (± 1.19)	85.73 (± 1.42)	71.51 (± 1.23)	37.91 (± 0.66)	31.46 (± 1.04)
Vitisin B	100.00 (± 1.07)	110.02 (± 2.84)	68.73 (± 1.18)	154.26 (± 1.03)	156.42 (± 1.10)
Peonidin 3-(6″-acetyl)-glucoside	100.00 (± 1.05)	81.07 (± 1.66)	46.73 (± 0.76)	43.34 (± 0.40)	15.20 (± 0.53)
Malvidin 3-(6″-acetyl)-glucoside	100.00 (± 1.01)	78.11 (± 2.75)	38.63 (± 0.45)	45.34 (± 0.74)	15.69 (± 0.09)
Delphinidin 3-(6″-coumaroyl)-glucoside	100.00 (± 1.23)	75.54 (± 1.17)	14.38 (± 0.27)	11.96 (± 0.22)	2.06 (± 0.06)
Malvidin 3-(6″-caffeoyl)-glucoside	100.00 (± 1.24)	78.60 (± 2.35)	15.90 (± 0.16)	11.42 (± 0.15)	6.37 (± 0.11)
Petunidin 3-(6″-coumaroyl)-glucoside	100.00 (± 1.06)	76.09 (± 1.90)	13.12 (± 0.15)	10.21 (± 0.11)	1.87 (± 0.06)
Peonidin 3-(6″-coumaroyl)-glucoside	100.00 (± 1.36)	125.09 (± 2.97)	14.33 (± 0.19)	10.44 (± 0.14)	2.75 (± 0.05)
Malvidin 3-(6″-coumaroyl)-glucoside	100.00 (± 1.00)	80.26 (± 1.65)	12.20 (± 0.27)	12.12 (± 0.04)	2.54 (± 0.02)
Mean	100	89	39	43	28

**Table 18 molecules-26-05454-t018:** Malvidin 3-glucoside absolute concentrations (µg/mL, mean ± SD) in the extracts.

	Extract A	Extract A50	Extract B	Extract C	Extract D
Malvidin 3-glucoside	4.97 (±0.04)	3.93 (±0.09)	2.07(±0.02)	2.71 (±0.03)	1.74 (±0.02)

**Table 19 molecules-26-05454-t019:** Antioxidant activity determined by the DPPH test. Values are reported as µg/mL in respect to the dry residue and to total polyphenol content.

Extract	IC_50_ µg/mL (Dry Residue)	IC_50_ µg/mL (Polyphenols)
Extract A	7.303 ± 0.338	2.062 ± 0.095
Extract B	38.667 ± 0.790	3.013 ± 0.062
Extract C	56.673 ± 3.087	3.847 ± 0.210
Extract D	52.000 ± 2.009	2.154 ± 0.083

## Data Availability

The data presented in this study are available on request from the corresponding author. The data are not publicly available.

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
