# Peer review of "Effect of Extraction Solvent and Temperature on Polyphenol Profiles, Antioxidant and Anti-Inflammatory Effects of Red Grape Skin By-Product"

_molecules, 2021, doi:10.3390/molecules26185454_

Round 1

Reviewer 1 Report

The study entitled „Effect of extraction solvent and temperature on polyphenol profiles, antioxidant and anti-inflammatory effects of red grape skin by-product” conducted by Baron et al. presents a detailed LC-MS qualitative profiling of red grape skin extracts obtained in different conditions:  EtOH-H2O 70:30; H2O at 100 °C -1 h + 24 hours at room temperature; H2O at 70 °C – 1h + 24 hours at room temperature; H2O at room temperature- 24h, followed by the assessment of the antioxidant and anti-inflammatory activity. The study was rigorously conducted, and the obtained results were thoroughly explained. However, it is suggested for the authors to add supplementary remarks in the manuscript to emphasize the originality of the article.

 Observations:

-  The novelty of this work should be emphasized since Vitis vinifera by-products were extensively studied up to date; a more detailed comparison with similar studies should be included.

- Please specify how the employed extraction conditions were selected? Did the authors previously investigate the thermal stability of polyphenols?

Reviewer 2 Report

The manuscript is well written, and the results are easy to follow. The manuscript clearly presents the results and comparisons. The interpretations and conclusions are proved by the data. I have only minor comments:

Please uniform the units in the whole text (mg/mL, μM).

Line 104. Please add information about the instrument used (model of LC-HRMS, company, country), analytical column and mobile phase used.

Line 216. Change "table 3" to "Table 3".

Lines 211, 215, 219, 221, 236, 243, 264, 275, 286, 296, 308, 333, 340, 366. Please remove "-"at the beginning of the sentence.

Line 219. Please change " LC-ESI-(-)/MS" to " LC-ESI-(-)-MS".

Line 221. Please change " Table 4." to " Table 4.". The same for "Table 10" (line 296) and "Table 12" (321).

Equation 3: What does mean "extract i"?

Table 10. What does mean "*"? Please explain.

Figure 7. I do not see marks "*", "**", "***" indicating statistical differences. I also suggest changing the color of points for experiments C or D.

Reviewer 3 Report

The manuscript titled “Effect of extraction solvent and temperature on polyphenol profiles, antioxidant and anti-inflammatory effects of red grape skin by-product” is related to recovery of polyphenols from red grape skin (by-product) with green solvents and analysis of obtained extracts in terms of polyphenol profile, antioxidant activity and anti-inflammatory activity. The strength of manuscript is usage of appropriate analytical methods however, experimental setup has major issues thus I would suggest rejection in its current form.

Major issues

It is clear that temperature has a positive effect on polyphenol recovery from grape skin. A mixture of ethanol and water had the highest concentration of polyphenols. Considering all, authors should perform also extraction with ETOH:H2O mixture at boiling temperature, since this setup would most likely result in even higher yield comparing to all other extracts (A-D). Without those results the manuscript is incomplete.

All compounds are semi-quantified with relative content. I could agree that standards are expensive and not even all existing but I personally believe that at least some standards could be used for quantification.

Minor issues

Page 2 Lines 90-95 - Please use either a); b) etc. or (extract A); (extract B) etc. there is a misture of 2 stilles

Page 4 Lines 165-166 – The second part of the sentence is confusing, please rephrase it.

Table 1 and Table 2 – Please use two decimals for percentage presentation

Table 2 – there are several numbers separated with a comma instead of a dot, please correct it

Page 5 Line 181 and Table 3 – please align retention time for tartaric acid

English and typos

Page 1 Line 46 - CAGR – Please give a full term

Page 2 – Line 73 – Please choose between “by” or “using”

Round 2

Reviewer 1 Report

The author has made a revision of the manuscript and the article was improved.

Reviewer 3 Report

The authors significantly improved the manuscript and it should be accepted after values for sample A50 are inserted in table 4.